# New Prospects on Neuroimaging in Von Hippel Lindau Disease—A Narrative Review

**DOI:** 10.3390/diagnostics14030309

**Published:** 2024-01-31

**Authors:** Nikodem Pietrzak, Katarzyna Jankowska, Oskar Rosiak, Wieslaw Konopka

**Affiliations:** Department of Otolaryngology, Polish Mother Memorial Hospital Research Institute in Lodz, 93-338 Lodz, Poland; nikodempietrzak@icloud.com (N.P.); k.jankowska52@gmail.com (K.J.); wieslaw.konopka@umed.lodz.pl (W.K.)

**Keywords:** von Hippel-Lindau syndrome, hemangioma, diagnostic imaging

## Abstract

(1) Background: Hemangiomas in Von Hippel-Lindau (VHL) syndrome patients are typically benign but pose threats due to their vital locations involving the central nervous system and the retina. An MRI is currently recommended as the gold standard for tumors associated with VHL in the head region. This narrative review aims to comprehensively outline current standards and recent findings related to imaging of retinal and CNS hemangiomas in Von Hippel-Lindau syndrome. (2) Material and Methods: A review in adherence to PRISMA guidelines using the search string “Von Hippel-Lindau hemangioblastoma imaging” was conducted on PUBMED and SCOPUS databases. (3) Results: After reviewing 455 titles and abstracts, 20 publications fulfilling the inclusion criteria were analyzed. The analysis included studies describing MRI, CT, optical coherence tomography, and PET/CT. (4) Conclusion: While MRI remains the gold standard for diagnosing head tumors in Von Hippel-Lindau syndrome, various PET/CT methods show promise as alternative imaging modalities.

## 1. Introduction

Von Hippel-Lindau syndrome (VHL) is a genetic disorder, most often inherited in an autosomal dominant manner, characterized by the formation of benign and malignant tumors in multiple locations. The most common manifestations of VHL disease include central nervous system (CNS) hemangiomas. Up to 72% of patients with VHL may have hemangioma in the cerebellum (16–69%), brainstem (5–22%), spinal cord (13–53%), caudate (11%), or supraspinatus (1–7%) [1]. Other locations of possible tumors include retinal hemangioblastomas (RH), renal clear cell carcinoma (RCC), pheochromocytomas (PPGL), pancreatic neuroendocrine tumors (PNET), and endolymphatic sac tumors (ELST)—a rare type of neuroectodermal tumor of the endolymphatic sac and duct [2]. The VHL Surveillance Guidelines Consortium and VHL Alliance have established consensus guidelines for the management of patients with von Hippel-Lindau (VHL) syndrome or at-risk (children of VHL patients) [3]. Table 1 presents a simplified scheme of screening tests.

Retinal hemangiomas (RH) are the first symptom in up to 77% of patients with VHL and may be the only sign of the disease [4]. They are benign tumors with vascular features that can lead to vision loss due to retinal exudation, fibrosis, vitreous and subretinal hemorrhage or retinal detachment. The most commonly observed manifestation of retinal hemangiomas (RH) in the context of von Hippel-Lindau (VHL) disease is predominantly in the form of peripheral retinal capillary hemangiomas (RCHs) [5]. These lesions, characterized by their distinctive vascular composition, primarily appear in the peripheral regions of the retina. However, it is notable that, albeit less frequently, hemangiomas can also exert a direct influence on the optic nerve head or affect the retinal area adjacent to the nerve, a condition referred to as juxtapapillary involvement. In such scenarios, the intrinsic vascular characteristics of these tumors, compounded by their anatomical proximity to critical ocular structures, notably increase the risk of impaired visual function, potentially leading to significant vision loss [6]. The early identification and accurate characterization of these hemangiomas are of paramount importance, not only due to the direct implications they hold for visual prognosis but also because of their considerable impact on the therapeutic approach [4,5,6]. Therefore, regular ophthalmologic evaluation should be conducted in patients with VHL, to identify and offer prompt treatment for new or active hemangioblastoma as early as possible [7]. All patients affected by VHL, and their at-risk relatives should undergo regular annual eye examinations starting in childhood, including visual acuity testing, fundus examination, fundus photography, and fluorescence angiography. The purpose of this diagnosis is to ensure early diagnosis and the implementation of prompt treatment [8,9].

Magnetic resonance imaging (MRI) is the preferred imaging modality for brain and whole nervous system tumor lesions in VHL syndrome, as it provides better imaging quality compared to computed tomography (CT) and other modalities. In addition, there are concerns about radiation exposure from repeated CT imaging in individuals with germinal tumor predisposition syndrome [10]. According to the current expert consensus of the International VHL Surveillance Guidelines Consortium, in asymptomatic patients with stable hemangiomas, an MRI with gadolinium contrast (Gd) of the craniospinal axis is recommended every 2 years or as soon as symptoms of the disease appear. Patients with progressive hemangiomas or enlarging peri-tumor cysts may require more frequent MRI to guide and/or modify treatment. The appearance of new CNS-related symptoms during the interval should prompt further MRI [3]. MRI-based screening and surveillance of asymptomatic patients should begin at age 11 and be completed in most patients by age 65 [11].

Diagnosis of hemangiomas in VHL syndrome remains a challenge. Patients undergo frequent, lengthy, and costly imaging procedures. The reported limitations and potential side effects of the diagnostic methods used today are prompting researchers to search for other methods of imaging tumor lesions in the course of the disease. An example of this is the discovery of gadolinium deposition in central nervous system tissues due to MRI screening for hereditary tumor syndromes, with currently unknown clinical implications, which has led researchers to consider limiting the use of this contrast agent [12]. The methods of PET/CT imaging in VHL syndrome are gaining popularity among researchers, as they minimize risk and enhance cost-effectiveness compared to conventional imaging methods.

The purpose of this study is to present the results of a narrative review of current standards and recent reports on the imaging diagnosis of retinal and CNS hemangiomas in patients with VHL syndrome.

## 2. Materials and Methods

A narrative review was performed using the “PRISMA” criteria [13]. A review of the scientific literature was performed on 31 October 2023, using online databases: PUBMED and Scopus. “von Hippel-Lindau hemangioblastoma imaging” was formulated as a research question for the review. Overall, 258 records from the PUBMED database, and 401 records from the SCOPUS database were obtained. In total, 204 duplicates were removed.

Two authors independently reviewed 455 titles and abstracts selected in stage one. The inclusion criteria were: Polish or English language of the study, studies on the methods, results and evaluation of imaging diagnostic (MRI, CT, OCT, PET/CT) of hemangiomas due to VHL syndrome. Publications in languages other than Polish or English, older than 10 years, and in the form of conference proceedings or letters to the editor were considered as exclusion criteria.

For further review, we included 16 studies that met the mentioned criteria and 4 studies from the citation review—Figure 1.

## 3. Results/Discussion

Imaging methods commonly used in screening and monitoring of VHL-associated tumors include computed tomography (CT) and magnetic resonance imaging (MRI). Contrast-enhanced CT is used to assess pancreatic and renal tumors in VHL; however, due to the risk of secondary malignancies from ionizing radiation exposure, it is not recommended as the first-line standard procedure, especially in patients with an increased risk of malignancies due to VHL. CT is also used to evaluate bone destruction in ELST tumors for surgical planning. CT is not recommended for screening and imaging of hemangiomas in the central nervous system and retina due to VHL [3]. There are sporadic reports of CT use in cases of subarachnoid hemorrhage secondary to hemangioblastomas in VHL [14]. There are also reports describing the imaging of retinal hemangioblastomas in VHL using CT; for example, Tang X et al. described punctate calcification lesions with soft-tissue density in the posterior part of the eyeball [15] as seen in Figure 2.

In individuals at risk of VHL syndrome who report ocular symptoms, optical coherence tomography (OCT) and OCT angiography are used. These methods are applied in the diagnosis and detection of retinal hemangioblastomas in the course of the disease. Patients with VHL disease may present with two distinct categories of retinal hemangioblastomas (RHs): the traditional nodular type and an unconventional flat type, tumors with reduced vascularity that pose a particular challenge for detection through clinical examination of the fundus [16]. Pilotto E. et al. report that the evaluation of the thickened peripapillary retinal nerve fiber layer correlates with the occurrence of retinal RH, potentially through pathological proliferation. They also note that in VHL patients without retinal hemangioblastomas, this layer is often thin [17]. Another publication on the application of OCT, suggests the potential role of hyper-reflective retinal foci as a possible biomarker for microglial activation due to VHL. In this study, the authors explore the potential pathogenesis for the development of retinal hemangioblastomas through inflammatory processes [18].

Figure 2 Retinal hemangioblastoma.

Magnetic resonance imaging of the craniospinal axis serves as the gold standard diagnostic method in the imaging of CNS hemangioblastomas [3]. Researchers describe lesions within the central nervous system as characterized by hypo- or iso- intensity of the perilesional nodule along with an increased intense signal of cerebrospinal fluid in the vicinity of the lesion (Figure 3). With the administration of gadolinium contrast (Gd), the image of the perilesional nodule in tumor-related changes is enhanced. In T2 imaging, the cyst is visualized as a hypointense structure [11,19,20]. RHs are observed as hyperintensities with significant enhancements in T1 with calcification, which may occur as the disease progresses.

Imaging patients with cochlear implants due to deafness resulting from damage to the inner ear caused by ELSTs remains a challenge. Recent advancements in cochlear implant technology made these devices MRI-compatible for up to 3 Tesla; however, if the suspected lesion is on the implanted side a significant artifact of up to 6 cm obscures the field of view. Thus, an alternative imaging method or magnet removal should be scheduled [21].

Since the discovery of gadolinium (Gd) deposition in the central nervous system, there have been considerations to minimize its usage [12]. Vanbinst AM et al. have proposed a shortened 35 min whole-body MRI protocol with only one injection of gadolinium (Gd), making VHL screening potentially faster, more cost-effective, and more convenient for patients [22]. However, there is currently a lack of data from other centers regarding the application and evaluation of whole-body MRI screening in individuals with VHL disease.

The limitations and associated risks of classical imaging methods are driving scientists to explore alternative imaging techniques for hemangiomas in the context of VHL syndrome. It has been observed that hemangioblastomas occurring in VHL stem from hematopoietic precursors, and hematopoietic stem cells express somatostatin receptors, similar to PNETs. Studies using positron emission tomography (PET) imaging with DOTA-TATE in VHL patients confirm previous reports of the avidity of these tumors to somatostatin analogs [23]. There are reports of the utility of 68Gallium [68Ga]-DOTA peptide (DOTATATE, DOTANOC, DOTATOC) PET/CT for their detection, a method that leverages the high expression of somatostatin receptors (SSTR) in neuroendocrine cells. Increased accumulation of markers within the tumor tissue is described (Figure 3) [23]. This method is currently used in the diagnosis of, among other things, pancreatic tumors over the course of VHL syndrome [3,24,25].

Shamim S. et al. propose 68Ga-DOTANOC PET/CT as a useful method for screening and monitoring accompanying tumors in patients with a germline mutation of the VHL gene. In their study, out of the 67 (17 hemangioblastomas of CNS, spine or retina) VHL-related lesions identified in the PET/CT scan, only 28 (42%) were previously detected using conventional imaging modalities. 68Ga-DOTANOC PET/CT revealed an additional 39 (58%) lesions, influencing patient management. According to the authors, this imaging method could be employed as a comprehensive imaging approach for VHL patients, serving as an alternative to currently recommended radiological examinations such as MRI [26]. 

Shell J et al. conducted a study comparing the clinical utility of 68Ga-DOTATATE PET/CT in detecting pancreatic neuroendocrine tumors (PNET) in patients with VHL and other VHL-related lesions, in comparison to the currently recommended imaging methods—CT/MRI. The study found that 68Ga-DOTATATE PET/CT detected more lesions overall compared to both CT and MRI. A total of 208 lesions were detected by CT in 66 scans, 94 lesions were detected by MRI in 33 scans, and 206 lesions were detected by 68Ga-DOTATATE PET/CT in 61 scans, with 24 lesions not described in CT or MRI studies. The authors also declare that an additional advantage of 68Ga-DOTATATE PET/CT is that it is non-nephrotoxic, unlike intravenous contrast agents used in CT, which is important for VHL patients due to the frequent radiological examinations they undergo throughout their lives. Many of these patients may develop chronic kidney disease after prior nephrectomies due to clear cell renal cell carcinoma [27].

Considering the fact that PET/CT imaging results in lower patient radiation exposure (5–7 mSv depending on the type of study) compared to conventional CT (8–24 mSv depending on the area and use of contrast), this could potentially be a sensible choice for screening and monitoring tumors associated with VHL [27]. Performing CT screening on individuals with von Hippel-Lindau syndrome can lead to significant radiation exposure, even when using dual-energy virtual non-contrast CT, potentially increasing the risk of secondary malignancies in patients with Von Hippel-Lindau syndrome [28]. It is also worth mentioning again that the recommended contrast-enhanced MRI in VHL patients is associated with the risk of gadolinium accumulation in the central nervous system, the consequences of which remain unknown. The potential use of PET/CT could possibly address this dilemma.

Considering the application of PET/CT methods in diagnosing PNET tumors in patients with VHL syndrome and the similarity of these tumors to others in the course of the disease, such as retinal hemangiomas, few authors have also described individual cases using different PET/CT methods in imaging hemangiomas in patients with VHL. In a case presented by Papadakis GZ et al., authors described the use of 18F-FDG and 68Ga-DOTATATE PET/CT in VHL Disease-Associated RH. The described tumor showed low-level F-FDG and increased Ga-DOTATATE activity, suggesting cell-surface overexpression of somatostatin receptors [29]. The case presented by Dev ID et al. described a symptomatic sporadic cerebellar hemangioblastoma attributed to VHL syndrome, which mimicked a meningioma on MRI and 68Ga-DOTANOC PET imaging [30].

The retrospective study presented by Oosting SF et al. on the application of (89)Zr-bevacizumab PET in the diagnosis of tumors in VHL, based on a group of patients with previously confirmed hemangiomas, emphasizes that VHL manifestations can be visualized using (89)Zr-bevacizumab PET, albeit with significant variation in marker accumulation. The researchers also suggest that (89)Zr-bevacizumab uptake does not predict disease progression within 12 months. (89)Zr-bevacizumab PET might offer a tool to select VHL patients for anti-VEGF therapy [31].

## 4. Conclusions

The MRI remains the gold standard for hemangiomas in the central nervous system and retina, with confirmed utility and high sensitivity and specificity based on numerous patient studies. However, this method has its limitations, as in the case of imaging patients with cochlear implants, as well as drawbacks: lengthy, costly, and often repeated procedures. There are prospects for shortening the screening protocol using MRI in patients experiencing VHL syndrome. Computed tomography carries the risk of patient radiation exposure and potential secondary carcinogenesis, especially in the case of frequently repeated examinations in patients with VHL syndrome. Promising imaging methods for head lesions include various methods of PET/CT imaging. Researchers emphasize their promising utility, potential multifunctionality, and minimization of side effects compared to traditional imaging studies. However, these methods require further evaluation, and additional studies are needed to confirm the reported findings.

## Figures and Tables

**Figure 1 diagnostics-14-00309-f001:**
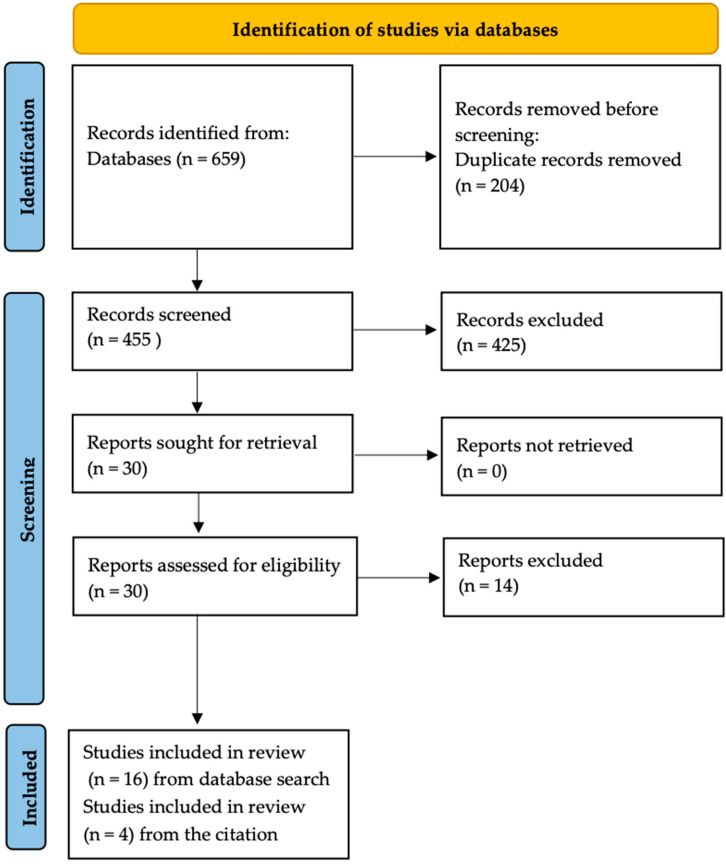
“PRISMA” Flowchart describing the process of literature review.

**Figure 2 diagnostics-14-00309-f002:**
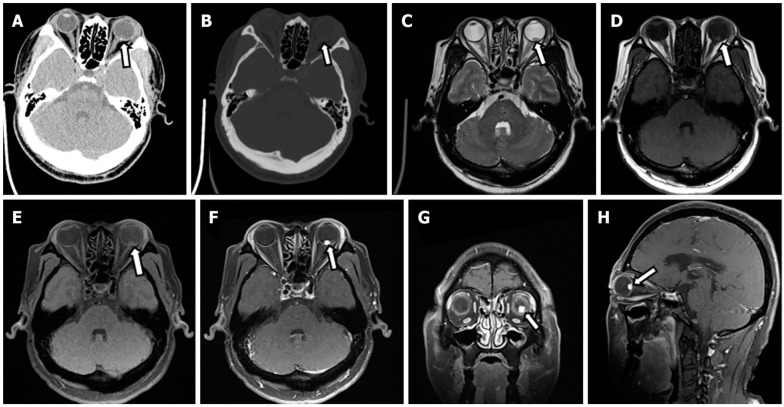
Computed tomography and magnetic resonance imaging of left retinal hemangioblastoma. (**A**): Computed tomography (CT) transverse soft tissue window of orbit showed punctate calcification on the posterior wall of the left eye ring and small patchy soft tissue density in the posterior part of the eyeball. The lesion measured about 5 mm × 8 mm, with an ill-defined border; (**B**): CT transverse bone window of orbital showed no obvious abnormal change of orbital bone; (**C**): The lesion was hypointense on transaxial T2-weighted sequence; (**D**,**E**): The lesion was slightly hyperintense on transaxial T1-weighted images (**D**) and transaxial T1-weighted + fat-suppression images (**E**); (**F**–**H**): Left posterior para-bulbar lesions were significantly enhanced on gadolinium-enhanced T1-weighted + fat-suppression images (White arrows represent lesion). Reprinted from Tang et al. [15].

**Figure 3 diagnostics-14-00309-f003:**
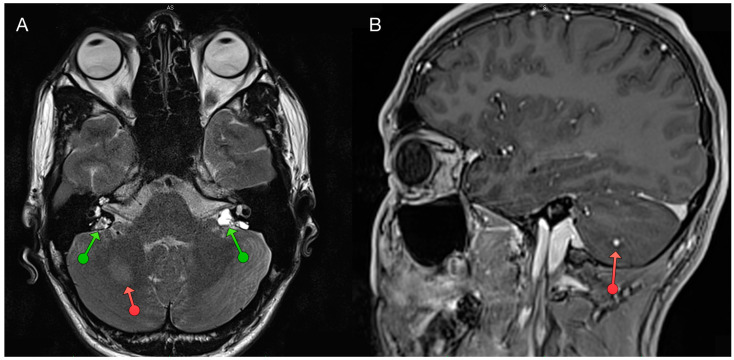
Magnetic resonance of a cerebellar haemangioblastoma and bilateral endolymphatic sac tumors (Author’s own material). (**A**) The T2 sequence shows an edema of tissue surrounding the hemangioblastoma (marked with orange arrow), medially and posteriorly to the vestibule two hyperintense tumors with typical “salt and pepper” appearance (green arrows). The tumors were post-operatively confirmed in histopathology as endolyphatic sac tumors in a von Hippel-Lindau patient. (**B**) The T1 sequence reveals a hyperintense hemangioblastoma located in the right cerebellum with a typical nodular appearence (orange arrow).

**Table 1 diagnostics-14-00309-t001:** A simplified scheme of screening tests in VHL patients.

Guidelines for Screening Procedures of Patients with/at Risk of VHL [3]		
Tumor	Age/Clinical Status	Methods/Procedures
All patients from birth		The most important element of patient screening is a thorough medical history and physical examination; at least annually
CNS (Hemangiomas)	11–65 years ^1^	MRI with and without gadolinium-based contrast agents every 2 years; new symptoms should trigger immediate MRI
	Pregnancy	MRI before pregnancy; Routine imaging without contrast agent; new symptoms should trigger MRI with contrast agents
Eye (RH)	<1–30 years	Binocular funduscopic examination (DFE) every 6–12 months
	>30 years	DFE annually
	Pregnancy	DFE before pregnancy
Kidney (RCC)	15–65 years ^1^	MRI of the abdomen with and without gadolinium-based contrast agents; Every two years if no tumor is found; Every 3–6 months if small (<3 cm) renal masses; Referral to a urologist if renal mass >3 cm found
	Pregnancy	MRI before pregnancy; if required during pregnancy avoid gadolinium-based contrast agents
Pancreas (PNET)	15–65 years ^1^	MRI with and without gadolinium-based contrast agents; Every 2 years, if small (<15 mm) or stable PNET; Referral to an expert in the management of PNETs if PNET >15 mm or enlargement
	Pregnancy	MRI before pregnancy; if required during pregnancy avoid gadolinium-based contrast agents
Ear (ELST)	11–65 years ^1^	Audiometric hearing test every 2 years
	15–20 years	Single baseline high-resolution MRI of the cerebellopotine angle and inner ear with and without contrast
Endocrine (PPGL)	At age 2 (symptomatic patients)	Clinical screening, blood-pressure monitoring
	5–65 years ^1^	Annual biochemical screening—plasma-free metanephrines

^1^ Stopping the screening procedure at age 65 if the patient has not developed lesions.

## Data Availability

The original contributions presented in the study are included in the article/; further inquiries can be directed to the corresponding author/s.

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
