# Peer review of "New Prospects on Neuroimaging in Von Hippel Lindau Disease—A Narrative Review"

_diagnostics, 2024, doi:10.3390/diagnostics14030309_

Round 1

Reviewer 1 Report

Comments and Suggestions for Authors

1.FIG 2 should be moved to the front of the reference section of the article

2.FIG 1 could be bigger and clearer

3.More references can be provided

4.The abstract section could be slightly modified to arouse reader interest

Comments on the Quality of English Language

good

Author Response

Dear Reviewer,

Thank you for taking the time and effort to review our work.

Response Ad 1 and 2. Figures were updated and figure quality was enhanced according to the reviewer's comments. 

Response Ad 3. The references provided have been obtained through a systematic review of the literature following the PRISMA guidelines, we have modified the references in the revised version. 

Response Ad 4. We have modified the abstract, please find the improved abstract section in the revised manuscript.

Reviewer 2 Report

Comments and Suggestions for Authors

Dear authors,  

the paper addresses an interesting topic but requires some revisions.  The title indicates the study design. The abstract provides an informative and summary of the study but I don’t understand why you only refer to hemangiomas at the lines 18-19 “While MRI remains the gold standard for diagnosing head hemangiomas in Von Hippel-Lindau syndrome, various PET/CT methods show promise as alternative imaging modalities”.   

Images are poor. You wrote about ocular diseases and hemangiomas. I think it is necessary to insert two images, one of the ocular pathology and another one of hemangiomas.  

- Figure 2. The images of haemangioblastoma needs a post-contrast T1w to demonstrate the mural nodule vividly enhances of the lesion.  

-  Lines 132-133: " [..] which were confirmed as endolymphatic sac tumors in a Von- Hippel Lindau patient”. Is the histology confirmed by a biopsy? The biopsy also confirmed the cerebellar hemangioblastoma or is it a diagnostic hypothesis?  Please improve these points.  

- Lines 211-222: the MRI remains the preferred diagnostic tool for hemangiomas in the central nervous system but I don’t understand in the conclusion the reason why the PET CT will be preferred. Like computer tomography (CT), PET generates its images with the help of a small dose of radioactive radiation. Also, the cochlear implant also creates artifacts in the PET. The MRI remains the method with the high sensitivity and specificity in detecting brain and orbital tumors, not only for the hemangiomas. Please improve the conclusion.

Comments on the Quality of English Language

Minor editing of English language required.

Author Response

Dear Reviewer, 

Thank you for taking the time to review our work, please find a point-by-point response below. 

1. "Dear authors,  

the paper addresses an interesting topic but requires some revisions.  The title indicates the study design. The abstract provides an informative and summary of the study but I don’t understand why you only refer to hemangiomas in the lines 18-19 “While MRI remains the gold standard for diagnosing head hemangiomas in Von Hippel-Lindau syndrome, various PET/CT methods show promise as alternative imaging modalities”.   

Response ad 1 The abstract was modified, we have changed this to “tumors” in von Hippel Lindau to include other neoplasms associated with this syndrome.:

2. Images are poor. You wrote about ocular diseases and hemangiomas. I think it is necessary to insert two images, one of the ocular pathology and another one of hemangiomas.  

- Figure 2. The images of haemangioblastoma needs a post-contrast T1w to demonstrate the mural nodule vividly enhances of the lesion.  

Response ad 2: Thank you for bringing this to our attention. We have updated figures, the heamangioma is shown in T1W to demonstrate mural nodule, and FLAIR to demonstrate perinodular edema. We have also included a retinal tumor in a separate figure in accordance with reviewers' suggestions which is cited from literature under Creative Commons 4.0 license. 

3.  Lines 132-133: " [..] which were confirmed as endolymphatic sac tumors in a Von- Hippel Lindau patient”. Is the histology confirmed by a biopsy? The biopsy also confirmed the cerebellar hemangioblastoma or is it a diagnostic hypothesis?  Please improve these points.  

Response ad 3: The endolymphatic sac tumor was confirmed by post-operative histopathological examination, the cerebellar hemangioblastoma is a diagnostic hypothesis, however, this patient was also confirmed by genetic testing to be suffering from von Hippel-Lindau syndrome therefore following clinical symptoms, genetic testing and endolymphatic sac tumor this diagnosis is almost certain. We have clarified that in the figure reference.

-4. Lines 211-222: the MRI remains the preferred diagnostic tool for hemangiomas in the central nervous system but I don’t understand in the conclusion the reason why the PET CT will be preferred. Like computer tomography (CT), PET generates its images with the help of a small dose of radioactive radiation. Also, the cochlear implant also creates artifacts in the PET. The MRI remains the method with the high sensitivity and specificity in detecting brain and orbital tumors, not only for the hemangiomas. Please improve the conclusion.

Response ad 4: We do not state that PET/CT will be a preferred diagnostic tool, only that it remains an option and requires further investigation. In cases where frequent MRI should be avoided or in a patient with a history of allergic reaction to gadolinium it could be an option. We have changed the opening sentence to avoid a misinterpretation.

From “Current guidelines recommend MRI as the preferred diagnostic tool for hemangiomas in the central nervous system and retina, with confirmed utility based on numerous patient studies.”

To “The MRI remains the golden standard for hemangiomas in the central nervous system and retina, with confirmed utility and high sensitifity and specificity based on numerous patient studies.”

Reviewer 3 Report

Comments and Suggestions for Authors

This is an interesting review regarding the different imaging modalities in VHL syndrome. 

As a Neurosurgeon I am used to MRI and sometimes CT to diagnose and follow my patients. The authors present the difficulties with CT and MRI and show that other options are possible for different indications. 

The information is important.

Different medical fields will be interested in different parts of the review. CNS, eyes and retina, kidneys, pancreas etc.  

Author Response

Dear Reviewer,

Thank you for taking the time and effort to review our work.

Reviewer 4 Report

Comments and Suggestions for Authors

The authors nicely reviewed imaging techniques for VHL in Polish and English-language literature. 

In the introduction: (Page 2, line 38) Retinal hemangiomas are described, however, the authors should mention the different types of these lesions depending on their location. The most common ocular lesions are peripheral RCHs. Less frequently, hemangioblastoma can affect the ON head itself or the retina adjacent to the nerve (juxtapapillary). This is essential since it has a major impact on the management and outcome. They should refer to the following paper and add it to the References: 

Al-Essa RS, Helmi HA, Alkatan HM, Maktabi AMY. Juxtapapillary retinal capillary hemangioma: A clinical and histopathological case report. Int J Surg Case Rep. 2021 Feb;79:227-230. doi: 10.1016/j.ijscr.2021.01.014. Epub 2021 Jan 15. PMID: 33485171; PMCID: PMC7820296.

Author Response

Dear Reviewer,

Thank you for taking the time and effort to review our work.

1. In the introduction: (Page 2, line 38) Retinal hemangiomas are described, however, the authors should mention the different types of these lesions depending on their location. The most common ocular lesions are peripheral RCHs. Less frequently, hemangioblastoma can affect the ON head itself or the retina adjacent to the nerve (juxtapapillary). This is essential since it has a major impact on the management and outcome. They should refer to the following paper and add it to the References: 

Al-Essa RS, Helmi HA, Alkatan HM, Maktabi AMY. Juxtapapillary retinal capillary hemangioma: A clinical and histopathological case report. Int J Surg Case Rep. 2021 Feb;79:227-230. doi: 10.1016/j.ijscr.2021.01.014. Epub 2021 Jan 15. PMID: 33485171; PMCID: PMC7820296.

 Response: We have enhanced the introductory section pertaining to retinal hemangiomas in alignment with the critical guidance provided by the reviewer. Additionally, we have updated the literature references to include the source recommended by the reviewer.  We also include a new Figure 2. demonstrating retinal hemangioma in line with other Reviewer's suggestions. Below is the appended introduction:

"The most commonly observed manifestation of retinal hemangiomas (RH) in the context of von Hippel-Lindau (VHL) disease is predominantly in the form of peripheral retinal capillary hemangiomas (RCHs) [5]. These lesions, characterized by their distinctive vascular composition, primarily appear in the peripheral regions of the retina. However, it is notable that, albeit less frequently, hemangiomas can also exert a direct influence on the optic nerve head or affect the retinal area adjacent to the nerve, a condition referred to as juxtapapillary involvement. In such scenarios, the intrinsic vascular characteristics of these tumors, compounded by their anatomical proximity to critical ocular structures, notably increase the risk of impaired visual function, potentially leading to significant vision loss [6]. The early identification and accurate characterization of these hemangiomas are of paramount importance, not only due to the direct implications they hold for visual prognosis but also because of their considerable impact on the therapeutic approach [4-6]."